# CRISPR/Cas9 screening using unique molecular identifiers

Bernhard Schmierer[1,†] (ID), Sandeep K Botla[1,†] (ID), Jilin Zhang[1] (ID), Mikko Turunen[2], Teemu Kivioja[2] (ID) & Jussi Taipale[1,2,*] (ID)

## Abstract

**Loss-of-function screening by CRISPR/Cas9 gene knockout with pooled, lentiviral guide libraries is a widely applicable method for systematic identification of genes contributing to diverse cellular phenotypes. Here, Random Sequence Labels (RSLs) are incorporated into the guide library, which act as unique molecular identifiers (UMIs) to allow massively parallel lineage tracing and lineage dropout screening. RSLs greatly improve the reproducibility of results by increasing both the precision and the accuracy of screens. They reduce the number of cells needed to reach a set statistical power, or allow a more robust screen using the same number of cells.**

**Keywords** CRISPR/Cas; genetic screening; massively parallel lineage tracing; unique molecular identifiers
**Subject Categories** Genome-Scale & Integrative Biology; Methods & Resources
**Mol Syst Biol. (2017) 13: 945**

## Introduction

Pooled CRISPR/Cas9 loss-of-function screening is a powerful approach to identify genes contributing to a wide range of phenotypes (Shalem *et al*, 2015). A library of guide sequences is integrated lentivirally into Cas9-expressing cells, which are then subjected to a selection pressure. Relative guide frequencies in the population before and after selection are quantified by next-generation sequencing (NGS) to determine both depleted and enriched guides.

The approach has been applied successfully (Gilbert *et al*, 2014; Koike-Yusa *et al*, 2014; Shalem *et al*, 2014; Wang *et al*, 2015), but suffers from several shortcomings: First, the presence of a guide does not necessarily cause loss of the corresponding gene, and cells sharing the same guide have distinct genotypes and phenotypes. Second, identification of guides that are under negative selection can be confounded by random drift and undersampling. Third,

growth characteristics of individual cells can vary substantially (Levy *et al*, 2015; Sandler *et al*, 2015) and the site of viral integration can affect the phenotype. For these reasons, each guide needs to be present in a large number of cells. In conventional screens, only the sum of all cells with a specific guide is measured, and no information regarding the distribution of cell behaviors can be obtained. Optimal identification of hit genes would require a method that individually tracks clonal lineages derived from single virus-transduced cells.

## Results and Discussion

Here, we address these issues by incorporating an RSL into the guide-library plasmid (Fig 1A) to allow tracing of hundreds of individual virus-transduced cell lineages in a CRISPR screen. In contrast to the use of barcodes in single-cell transcriptome analysis following CRISPR/Cas9 gene editing (Adamson *et al*, 2016; Dixit *et al*, 2016; Datlinger *et al*, 2017), we use unique molecular identifiers (Kivioja *et al*, 2012 and references therein) to either trace single clones (Kalhor *et al*, 2017) of identically edited cells, or very small pools of sublineages composed of cells with different editing outcomes at the same locus (Fig 1B). Such massively parallel lineage tracing enables both lineage dropout analysis (LDA), and the creation and analysis of internal replicates (IRA), while retaining the option of conventional, total read count analysis (TCA, Fig 1C).

To demonstrate the power and flexibility of the approach, we screened the human colorectal carcinoma cell line RKO for essential genes with an RSL-guide library targeting 2,325 genes with 10 guides per gene (Wang *et al*, 2015). Briefly, Cas9-expressing RKO cells were transduced with the lentiviral guide library, and samples were taken at Day 4 and Day 28 after transduction (control and treatment time points, respectively). Guide frequencies in the two time points were then assessed by NGS. The experiments were run in duplicate and at far larger screen size (we define "screen size" as the number of cells per guide sequence) and sequencing depth (reads per guide) than previous screens (Shalem *et al*, 2014; Wang *et al*, 2015). Such redundancy allows subsequent subsampling using the RSL information, and robust testing of different analytical

1 Department of Medical Biochemistry and Biophysics, Karolinska Institutet, Stockholm, Sweden
2 Genome-Scale Biology Research Program, Faculty of Medicine, University of Helsinki, Helsinki, Finland
*Corresponding author. Tel: +46 8 585 86895; E-mail: jussi.taipale@ki.se
†These authors contributed equally to this work

**A** Library design.

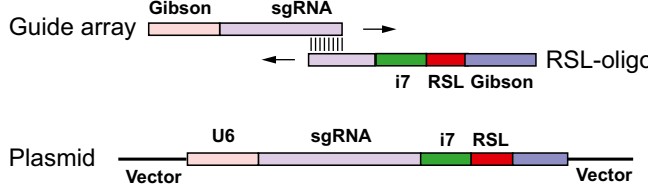

**Sequencing library**

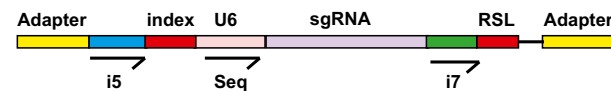

**B** Lineage drop out and lineage depletion

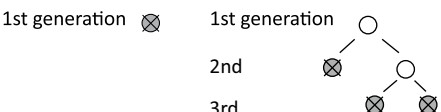

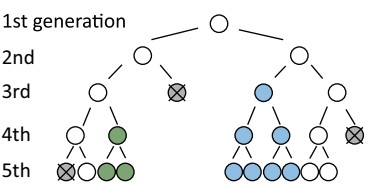

**C** Levels of analysis

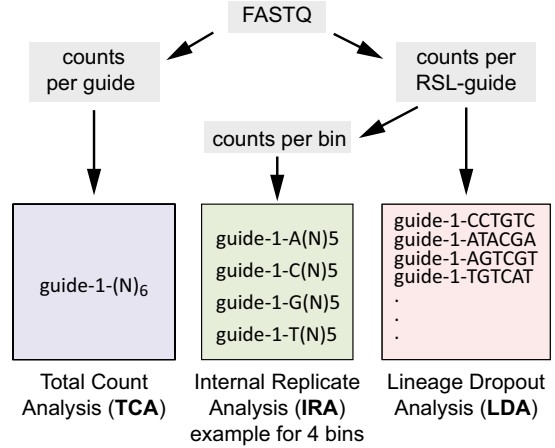

**D** Screen size and sequencing depth

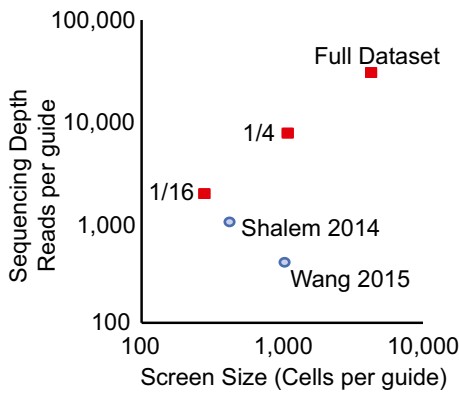

**Figure 1. CRISPR/Cas9 screening using unique molecular identifiers.**

A  Library design and cloning. Top: The guide library is synthesized as an oligonucleotide array; the RSL-part is synthesized as a single, overlapping oligonucleotide containing a 6-bp random sequence (RSL) and the Illumina index primer (i7) binding sequence. Guide-array and RSL-oligonucleotide are annealed and double-stranded. Homology arms for Gibson assembly are also indicated. Middle: Guide plasmid. The i7 index read primer binding site and the RSL are located downstream of the sgRNA termination signal and are not part of the guide RNA. Bottom: Sequencing library. Sequencing is performed using a custom primer (Seq) placed directly upstream of the guide (gRNA). The sample index and RSL are read as two index reads with Illumina i5 and i7 index primers, respectively (20 + 6 + 6 sequencing cycles).

B  Lineage dropout versus lineage depletion. Depending on the kinetics of editing, single cell lineages harboring a single RSL-guide against an essential gene can either disappear (dropout) or decrease in their abundancy (depletion). Top: Dropout happens if the editing occurs early on, either before the cell can divide, or in several independent events at later time points (gray, dead cell; white, unedited cell). Bottom: In lineage depletion, editing occurs either after several cell divisions and/or with several different outcomes (blue and green edits), some of which will retain gene function of the essential gene. In such cases, the traced lineage is comprised of several sublineages.

C  RSL-guides allow additional methods of analysis. In total count analysis (TCA, left), RSL information is ignored and only the sum of readcounts for all RSL-guides is taken into account. In internal replicate analysis (IRA, middle), readcounts of RSL-guides are binned such that internal replicates are created for each guide. The example shown bins RSL-guides into four internal replicates; however, RSL-guides can be binned in any number of replicates. In lineage dropout analysis (LDA, right), each RSL-guide is monitored separately.

D  Screen size and sequencing depth. The screens were performed at a very large screen size of roughly 4,500 cells per guide and sequenced to a depth of 30,000 reads per guide. Using RSL information, the data from these oversized experiments were then subsampled bioinformatically to approximately one quarter and 1/16, to test different analysis methods at different screen sizes. The corresponding values for two published screens are indicated for comparison (Shalem *et al*, 2014; Wang *et al*, 2015).

methods at varying screen sizes (Fig 1D). Perhaps counterintuitively, analysis of hundreds of RSL-labeled cell lineages per guide neither requires more cells per guide, nor markedly deeper sequencing, because any screen needs to use a relatively large number of cells per guide to achieve statistical power. Tagging each individual lineage incurs no cost. The RSL approach simply splits the total

guide read count obtained to read counts representing individual constituent cell lineages (Fig EV1), thus increasing the amount of information that is obtained, and consequently improving both precision and accuracy of the screen.

The plasmid library input contained 78 million unique RSL-guide combinations, 93 % of which were also detected in the virus-transduced

## A Internal replicate analysis (IRA) at the guide level

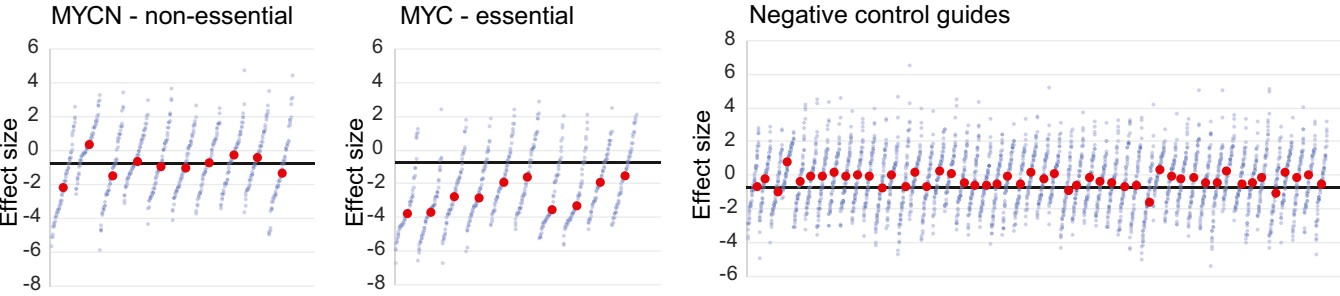

## B Internal replicate analysis (IRA) at the gene level

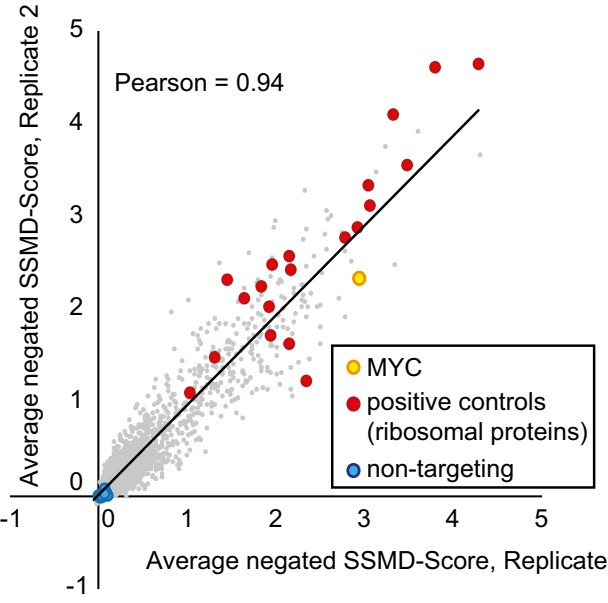

## C Lineage dropout analysis (LDA) - Replicate concordance

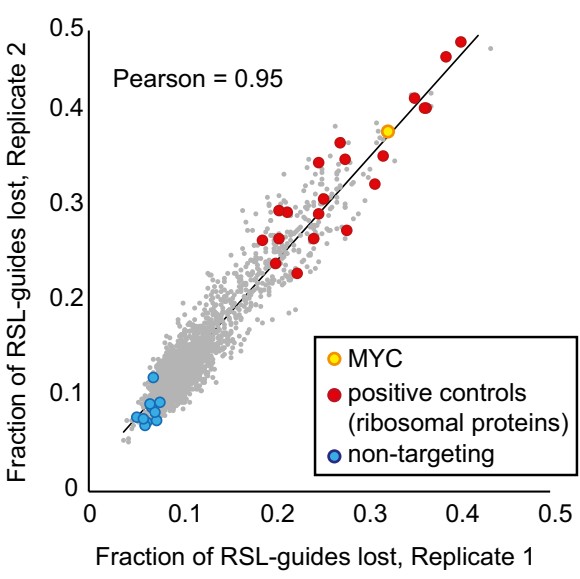

**Figure 2. RSLs enable internal replicate and lineage dropout analyses.**

A   Internal replicate analysis (IRA) at the guide level. RSL-guides were binned to create 64 internal replicates. Effect sizes (log2 fold change in readcount between Day 4 and Day 28 after virus transduction) for each bin are plotted in ascending order, 10 guides each for MYCN (top left) and MYC (top middle), as well as 50 representative, non-targeting guides (top right, these non-cutters seem to have a small fitness advantage). Red dots, median effect size (MES) of the 64 internal replicates (effect size of each internal replicate is one blue dot); black line, MES of all guides in the library. Hits for this type of data were called by SSMD score, see Materials and Methods for details. More examples are shown in Fig EV3.

B   Internal replicate analysis (IRA) at the gene level. RSL-guides were binned into 64 internal replicates. SSMD scores were calculated for each guide and averaged across all guides targeting the gene to obtain a score for each gene. For plotting, average SSMD scores for each gene were negated for easier comparison with Fig 1C. Red, positive controls (ribosomal proteins); blue non-targeting controls; orange, MYC; black line, linear regression.

C   Lineage dropout analysis (LDA). The fraction of RSL-guides lost from Day 4 to Day 28 in each experimental replicate is plotted for each gene (average over all guides targeting the gene). Red, positive controls (ribosomal proteins); blue, non-targeting controls; orange, MYC; black line, linear regression. The number of virus-transduced cell lineages lost is the most direct readout of the guide effect on cell viability.

cell populations (Fig EV2). Based on the Poisson distribution, this indicates that about half of the RSL-guides were incorporated into one or two cell lineages. Because only a subset of the cells can be harvested at each time point, undersampling is unavoidable, and some cell lineages (and corresponding RSL-guides) were present only in one of the time points (Venn diagram, Fig EV2). Such undersampling and loss of cell lineages occur whether or not RSLs are present, however go undetected in their absence. With RSLs, the effect becomes apparent and can be used in quality control of individual experiments as well as in filtering out inconsistently sampled lineages prior to data analysis.

RSL-labeled, distinguishable guide sequences can be used to split the data into internal replicates, which allow the usage of classical statistical tools to test for significant differences. To demonstrate the approach, RSL-guides were binned into 64 internal replicates per guide. The median effect size (Fig 2A) and a median-based version of strictly standardized mean difference (SSMD; Zhang, 2007) were then used to rank the guides (internal replicate analysis using SSMD, IRA/SSMD, Fig EV3). The average of all guide scores for each gene was used as a gene score (Fig 2B). The relatively high variability within internal replicates (Fig EV4) is consistent with

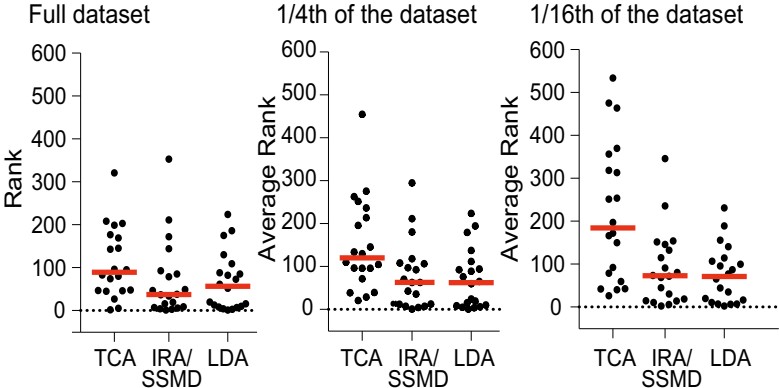

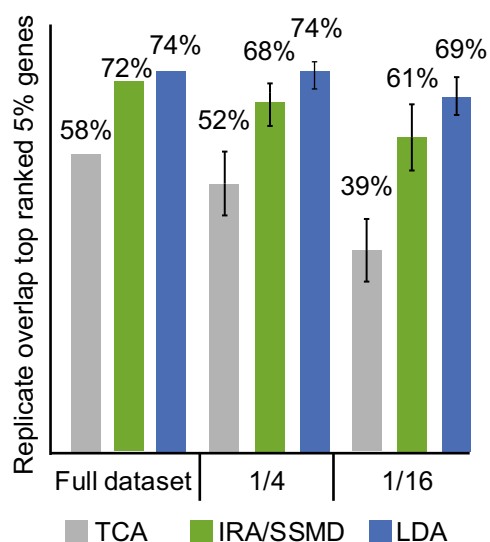

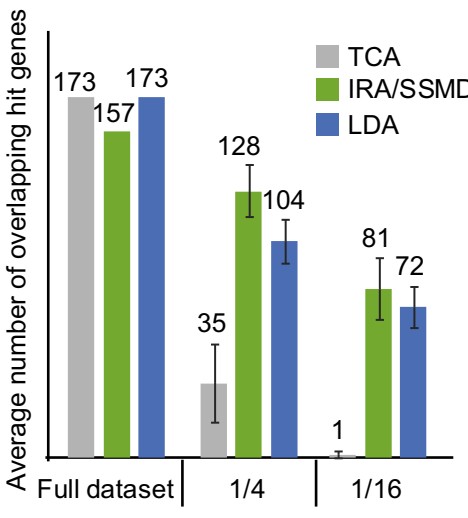

**Figure 3. RSLs improve precision and accuracy of hit calling.**

A   RSLs increase accuracy of hit calling. Ranks or average ranks of known positive controls (20 ribosomal proteins out of a total of 2,335 interrogated genes) in one experimental replicate for the full screen size (left, rank is plotted), as well as one quarter (middle) and 1/16 (right) of the full screen size (average rank of four subsamples is plotted). Red line, median rank. At all screen sizes, IRA/SSMD analysis and LDA assigned lower ranks to the positive controls than TCA. In TCA, the variance of the ranking increased substantially with decreasing screen size, but not in the two RSL-based methods, which are robust and allow hit calling from fewer cells.

B   RSLs increase the precision of gene ranking. Average percent overlap of the top-ranked 5% of genes (116 genes) between two experimental replicates. Error bars, standard deviation of four subsamples. LDA is the most precise method, followed by IRA/SSMD. Again, both RSL-based methods outperform TCA and are more robust at smaller screen sizes.

C   RSLs boost statistical power. Hit gene (FDR < 1%) overlap between experimental replicates at full screen size, one quarter, and 1/16 of the full screen size. Error bars, standard deviation for hit gene overlap between four subsamples in experimental replicate 1 and four subsamples in experimental replicate 2 (16 comparisons in total). Only at full screen size, TCA matches the RSL-based analyses. At more practical screen sizes, both RSL-based analyses have much higher statistical power and identify considerably more hit genes.

long culture time and the known variability of cell growth under culture conditions (Levy *et al*, 2015). In addition, variation in Cas9- and guide-RNA expression and distinct repair outcomes are expected to cause initial and long-term variation of growth characteristics of individual lineages. Such variability is present whether or not RSLs are included, however is not readily detected in the conventional total count analysis.

Finally, RSL-labeled guides enable lineage dropout screening, where gene hits are called solely based on the number of lost

RSL-guide lineages (lineage dropout analysis, LDA, Fig 2C). This is the simplest way of analyzing RSL data.

To evaluate IRA/SSMD and LDA, and to compare them with conventional TCA performed with the pipeline MAGeCK (Li *et al*, 2014), we assessed the ranks of a set of known essential genes (accuracy), and the hit gene overlap between experimental replicates (precision). In principle, RSL-based methods should outperform TCA when the number of cells per guide is relatively low, and their benefit should progressively decrease as the number of cells per

guide approaches infinity. Thus, the comparisons were performed using the complete dataset, and subsamples of the data that were similar in sample size to published screens (Shalem *et al*, 2014; Wang *et al*, 2015; Fig 3).

Both IRA/SSMD and LDA were more accurate than TCA, as indicated by lower hit ranks of 20 known essential, ribosomal proteins (Figs 3A and EV5). Both IRA/SSMD and LDA were also more precise than TCA, with much improved replicate concordance between the top-ranked 5% of genes (Fig 3B). Consistently with the theoretical considerations, our analysis revealed that the RSL-based methods were far more robust at practically used screen sizes when compared to TCA. The number of highly significant hit genes (as defined by a false discovery rate smaller than 1%) was massively increased in IRA/SSMD and lineage dropout analysis when compared to total read count analysis (Fig 3C). Only at dramatically exaggerated screen size, TCA performed comparably well. Thus, at practicable screen sizes (hundreds of cells per guide), RSL-based methods outperform TCA. The availability of two different RSL-based analysis methods provides increased flexibility; allowing the user to choose the most appropriate method for the specific design of a particular screen.

To summarize, RSLs dramatically improve accuracy, precision, and statistical power in CRISPR/Cas9 screening. The RSL strategy is not limited to CRISPR knockout screening, but can be applied in other screening methods such as CRISPR-dependent inhibition or activation screens (Gilbert *et al*, 2014; Konermann *et al*, 2015). We expect the RSL method to become instrumental in the interrogation of small genomic features, for example, exons, promoters, and even individual transcription factor binding sites. In many of these cases, there is just one possible guide sequence, and in such cases, the inclusion of RSLs is the only way to obtain the replicates that are required for hit calling. In the absence of precise knowledge of both on- and off-target activity, inclusion of multiple guide positions is, however, still important, and rescue experiments and/or analysis of the mutational spectrum of the cutsite are necessary to establish that the mutation induced by the guide results in the observed phenotype. Incorporation of RSLs is technically straightforward and does not require a higher number of cells or sequencing reads compared to conventional approaches. In contrast, RSLs give the same statistical power at a lower number of cells per guide, improving the economy of CRISPR/Cas9 screens. They also improve accuracy and precision at a given number of cells per guide, which is particularly advantageous in cases where cell numbers are limiting, such as in primary cells, or in very large genome-wide screens targeting genes or genomic regulatory regions.

# Materials and Methods

### Oligo nucleotide synthesis and library cloning

The guide library targets 2,325 genes and contains a total of 23,279 guides (Dataset EV1). The targeted gene set contains all human transcription factors (Vaquerizas *et al*, 2009), other genes of interest as well as ribosomal proteins as positive controls and 101 non-targeting guides as negative controls. All sgRNA sequences used in this library were taken from a previously published, genome-wide library (Wang *et al*, 2014). Oligos were synthesized on an array

(CustomArray). A single overlapping oligo containing random six base pairs as RSLs was annealed to the oligo library, and double-stranded to create the insert for cloning by Gibson assembly into the lentiviral vector pLenti-Puro-AU-flip-3xBsmBI, which was created by modifying lentiGuide-Puro (a gift from Feng Zhang, Addgene #52963) by replacing the sequence

```
gttttagagctagaaatagcaagttaaaa......TTTTTT with
gtttAagagctagaaatagcaagttTaaa......TTTTTTcgtctct
```

to create an AU-flip (Chen *et al*, 2013) and an additional BsmBI site downstream of the tracrRNA. The full insert sequence is

**ggctttatatatcttgtgtggaaaggacgaaacaccgnnnnnnnnnnnnnnnn** **nnnnngtttaagagctagaaatagcaagtttaaataa**ggctagtccgttat caacttgaaaaagtggcaccgagtcggtgctttttGATCGGAAGAGCAC ACGTCTGAACTCCAGTCACnnnnnnnaagcttggcgtaactagatcttgag acaaa

The fragment from the oligo array is shown in bold; the overlapping fragment containing the RSL and the Illumina i7 index primer (uppercase) was synthesized as a single 119-bp oligo (italics). This oligo was annealed to the oligo library (overlapping region bold italics) and double-stranded using outer primers (underlined).

### Gibson assembly, transformation, and amplification of the library

100 ng vector and 12 ng insert were assembled in a total reaction volume of 100 μl (NEBuilder® HiFi DNA Assembly Master Mix, NEB). The reaction was cleaned via a Minelute reaction cleanup column (Qiagen) and transformed into 6 × 50 μl electrocompetent *E. coli* (Endura™ ElectroCompetent Cells, Lucigen) using a 1.0 mm cuvette, 25 μF, 400 Ohms, 1,800 Volts. Bacteria were plated on several 24 × 24 cm agar plates, and colonies were grown overnight at 30°C. Colonies were scraped into LB medium, and the contained plasmids were isolated by Maxiprep.

### Library packaging

The library was packaged in HEK 293T cells by cotransfecting the library plasmid and the two packaging plasmids psPAX2 (a gift from Didier Trono, Addgene #12260) and pCMV-VSV-G (a gift from Bob Weinberg, Addgene #8454) in equimolar ratios. After 48 h, the virus-containing supernatant was concentrated 40-fold using Lenti-X concentrator (Clontech), aliquoted for one time use, and stored at −140°C.

### Cell lines and cell culture

RKO cells used in this study were purchased from ATCC. Cells were regularly tested for mycoplasma using the Mycoalert detection kit (Lonza; cat# LT07-218).

### Creating editing-proficient Cas9 cell lines

To rapidly generate editing-proficient cell lines, we synthesized a lentiviral construct (pLenti-Cas9-sgHPRT1) that encodes a codon optimized WT-SpCas9 that is flanked by two nuclear localization

signals (derived from lenti-dCAS-VP64_Blast, a gift from Feng Zhang, Addgene #61425). In addition, the construct codes for blasticidin resistance and carries an sgRNA against HPRT1 (GATGTGATGAAGGAGATGGG). HPRT1 loss confers resistance to the antimetabolite 6-thioguanine (6-TG). Lentivirally transduced cells were selected in 5 μg/ml blasticidin and after one week to 10 days additionally with 5 μg/ml 6-TG until control cells had died. Only cells that both express Cas9 and are editing proficient, as indicated by loss of HPRT1 function, will survive. The method allows rapid establishment of a pool of editing-proficient cells. Compared to single cell clones, this method retains the genetic heterogeneity of the original cell line, avoids potential clonal effects of the particular integration site of Cas9, and greatly accelerates cell line generation. These benefits need to be weighed carefully against possible disadvantages, such as synthetic lethality with HPRT1 loss, or potential effects of the presence of a second guide in the cell.

### Library transduction

Per experimental replicate, 100 million RKO Cas9 cells were transduced with the library virus. Two separate replicates were transduced. Cells were then selected for guide integration and expression by 1 μg/ml puromycin selection for 48 h. A proportion of cells will contain more than one guide. Because of the vast number of RSL-guides, any ineffective passenger guides will associate with effective guides randomly and will not be significantly enriched or depleted in the population.

### Cell propagation and sample preparation

Cells were kept in culture for a total of 28 days after transduction by sub-culturing them every 3–4 days. 100 million cells were reseeded at each split, and genomic DNA was prepared from at least 50 million cells at Day 4 and Day 28 after transduction. Day 4 after transduction was considered the control time point.

### Preparation of the sequencing library from genomic DNA

Genomic DNA was isolated using Blood and Tissue Maxi Kit (Qiagen), and 200 μg, theoretically corresponding to 30 million diploid cells, was used as PCR template in 40 parallel PCR1 reactions (5 μg template DNA each) using KAPA HiFi HotStart polymerase (KAPA Biosystems). After 14 cycles, the reactions were pooled. PCR2 used 5 μl of pooled PCR1 as template and was run for 19 cycles; PCR3 used 2 μl of PCR2 as template and was run for 14 cycles. The resulting product of 288 bp was gel purified and sequenced with a custom primer (CRISPRSeq) and the i5 and i7 index primers by running 20 + 6 + 6 cycles on the Illumina HiSeq4000, where i7 reads the RSL and i5 the illumina sample index.

Primers used for library preparation and sequencing:

| PCR1_FW | GGACTATCATATGCTTACCGTAACTTGAAAGTATTTCG |
|---|---|
| PCR1_REV | CTTTAGTTTGTATGTCTGTTGCTATTATGTCTACTATTCTTTCC |
| PCR2_FW | TCTTTCCCTACACGACGCTCTTCCGATCTCTTGTGGAAAGGACGAAACAC |

| PCR2_REV | AGAAGACGGCATACGAGATCTGCCATTTGTCTCAAGATCTAGTTAC |
|---|---|
| PCR3_FW | AATGATACGGCGACCACCGAGATCTACAC[i5]TCTTTCCCTACACGACGCTCTTCCG |
| PCR3_REV | CAAGCAGAAGACGGCATACGAGATCTGCCATTTG |
| CRIPSRSEQ | CGATCTCTTGTGGAAAGGACGAAACACCG |

**Final amplicon for sequencing (n indicates the guide, bold n represents the sample index, capital N the RSL, sequencing primer is underlined)**

aatgatacggcgaccaccgagatctacac**nnnnnn**tctttccctacacga cgctcttc<u>cgatctcttgtggaaaggacgaaacaccg</u>nnnnnnnnnnnnnn nnnnnnngtttaagagctagaaatagcaagtttaaataaggctagtccgtt atcaacttgaaaaagtggcaccgagtcggtgcttttttgatcggaagagca cacgtctgaactccagtcac*NNNNNN*aagcttggcgtaactagatcttgag acaaatggcagatctcgtatgccgtcttctgcttg

### Scripts used for counting RSL-guides

RSL-guides were counted in the original fastq files with the Perl scripts *Batch-GuideUMI-count-p0.1.pl*, which requires the script *GuideUMI-count-p0.1.pl*.

### Binning of RSL-guide counts for creation of internal replicates in IRA/SSMD analysis

Binning was done using the script *Bin-count-TruncatedUMIs.pl*. The script bins according to RSL sequences, taking the first base (4 bins), first two bases (16 bins), etc. into account. Generally, sequences whose sum of readcounts in control and treatment was less than five were filtered out prior to data analysis.

### IRA/SSMD analysis of read count data

Data were normalized to total read count: $c_{ij}$ and $t_{ij}$ represent the raw read counts for RSL-guide $j$ in guide-set $i$ for control (Day 4 after lentiviral transduction) and treatment (Day 28 after lentiviral transduction), respectively. The normalized read counts $c'_{ij}$ and $t'_{ij}$ are then

$$c'_{ij} = c_{ij} \frac{\sum_{ij}(c_{ij} + t_{ij})}{2 \sum_{ij} c_{ij}}$$

$$t'_{ij} = t_{ij} \frac{\sum_{ij}(c_{ij} + t_{ij})}{2 \sum_{ij} t_{ij}}$$

### Median effect size and variability of the guide-sets

We defined the effect size $ES_{ij}$ for each RSL-guide or bin $j$ in guide-set $i$ as the log2 of the fold change between treatment count and control count. To handle total loss of an RSL-guide or bin in the treatment sample, we added a pseudo-count of 1 to all counts:

$$ES_{ij} = log_2 \frac{t'_{ij} + 1}{c'_{ij} + 1}$$

Next, we calculated the median effect size for guide-set $i$, $MES_i$, and the median of the absolute deviations (MAD) of all RSL-guides or bins $j$ in guide-set $i$ from $MES_i$

$$MES_i = \underset{j}{median}\, ES_{ij}$$

$$MAD_i = 1.4826 \underset{j}{median}\, |ES_{ij} - MES_i|$$

The factor 1.4826 was chosen such that the MAD is approximately equal to the standard deviation under the assumption of normal distribution (Zhang, 2011).

### Median effect size and variability of the control guide-sets

The RSL library contains 101 non-targeting guide-sets. We calculate a single median effect size and MAD for this control set in the following way:

Median effect size of all non-targeting RSL-guides

$$MES_{CON} = \underset{ij}{median}\, ES_{ij}^{NONT}$$

Median absolute deviation of all non-targeting RSL-guides:

$$MAD_{CON} = 1.4826 \underset{ij}{median}\, \left| ES_{ij}^{NONT} - MES_{CON} \right|$$

### Strictly standardized mean difference

Strictly standardized mean difference is a measure for the significance of the difference in behavior of sample $i$ and the non-targeting controls. It takes into account both the effect size and the variability of the data.

$$SSMD_i = \frac{MES_i - MES_{CON}}{\sqrt{MAD_i^2 + MAD_{CON}^2}}$$

For samples with relatively small effect size, the SSMD can still become large if the spread is small. We thus introduce a score in which the effect size weighs more strongly, and which is used as a ranking parameter:

$$Score_i = MES_i\, |SSMD_i|$$

For hit calling, the average score and standard deviation were calculated for all non-targeting guide-sets.

The script used to do these calculations is *IRA-SSMD.sh*, which calls the script R-script *IRA-SSMD.R*. Guide-sets were then ranked according to their score. A gene hit list was obtained by analyzing the ranked guide list with the "pathway" function of MAGeCK, v0.5.6 (Kolde *et al*, 2012; Li *et al*, 2014b) using Dataset EV2.

### Lineage dropout

An RSL-guide was considered a dropout if it had less than two read-counts in the treatment time point. The numbers of RSLs per guide at Day 4 and Day 28 were then used to calculate an effect size (log2 fold change). Guides were ranked according to effect size, and significantly depleted genes were called with the "pathway" function of MAGeCK, v0.5.6 using Dataset EV2.

### Subsampling

For subsampling the full data set, RSL-guides were grouped according to their RSL-sequence. For medium screen size, the whole dataset was split into four groups (RSLs starting with A, C, G, and T). For small screen size, the whole dataset was split into 16 groups, the first four of which (AA, AC, AG, AT) were used for analysis. Such subsampling simulates both decreased sequencing depth and a smaller number of cells per guide (smaller screen size). Subsamples were used as replicates in the analyses shown in Fig 3.

### Data and software availability

Raw sequencing data: European Nucleotide Archive, PRJEB18436. Computer scripts: GitHub http://github.com/zhjilin/RSLC.

**Expanded View** for this article is available online.

### Acknowledgements

The authors would like to thank Drs Jenna Persson, Inderpreet Kaur Sur, and Minna Taipale for suggestions on the manuscript. Part of this work was carried out at Karolinska High Throughput Center (KHTC) and the High Throughput Genome Engineering Facility (HTGE) at Science for Life Laboratory, Sweden. This work was supported by the Academy of Finland, grant 250345; the Knut and Alice Wallenberg Foundation, grant KAW 2013.0088; and the Center for Innovative Medicine (CIMED) project grant "Growth Control and Cancer".

### Author contributions

BS, SKB, and JT developed the approach; BS, SKB, and MT performed the experiments; BS, SKB, JZ, and TK analyzed the data; BS and JT wrote the manuscript.

### Conflict of interest

The authors declare that they have no conflict of interest.

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
