## [Review Process File · Molecular Systems Biology]

CRISPR/Cas9 screening using unique molecular identifiers

Bernhard Schmierer, Sandeep K. Botla, Jilin Zhang, Mikko Turunen, Teemu Kivioja & Jussi Taipale

Corresponding author: Jussi Taipale, Karolinska Institutet & University of Helsinki

Review timeline:

Submission date:	24 June 2017
Editorial Decision:	24 August 2017
Revision received:	15 September 2017
Accepted:	18 September 2017

Editor: Thomas Lemberger

Transaction Report:

1st Editorial Decision

24 August 2017

Thank you again for submitting your work to Molecular Systems Biology and apologies for the delay in getting back to you which was caused by the difficulty finding reviewers during the summer break. We have now finally heard back from the two of the three referees who accepted to evaluate the study. Given that their recommendations are very similar, I prefer to make a decision now rather than delaying the process further.

As you will see, the referees find the topic of your study of potential interest and are rather positive. They raise however a series of concerns and make suggestions for modifications, which we would ask you to carefully address in a revision of the present work. This entails addressing their requests for clarification and more rigorous statistics. With regard to the presentation, you should also feel free to add a Figure if you feel it is necessary.

Please revise the manuscript accordingly and make sure to consult our instructions to authors, in particular for the formatting of "Expanded View Figures" (<http://msb.embopress.org/authorguide>).

With regard to providing the data and the computer code, we would kindly ask you to add a formal "Data and software availability" section after Materials & Methods.

REVIEWER REPORTS

Reviewer #1:

Schmierer et al. present a nice twist on CRISPR screens. Although previous work (e.g. Perturb-Seq papers from Regev and Weissman labs) already have incorporated UMIs in their pooled CRISPR

screens, I think a paper emphasizing the advantages of UMIs with a careful comparison to non-UMI screens will be useful for the field.

A few comments:

- Line 62 mentions binning of 64 replicates (barcodes per guide) but Fig. 1b indicates 4 bins under IRA.
- TracrRNA in Figure 1a is incorrect. Please re-label as sgRNA.
- It would be great to get a better sense of the variability between the IRA bins. Could the authors show data on the variability of these replicates in a few different example genes? And summarize the variability across all genes in the library?
- For Figure 1d, it is unclear what exactly is being plotted. Does each curved line contain 64 replicate bins? If so, the dots must be plotted with alpha shading. Otherwise, it is difficult to see where the bins concentrate. Also, based on these plots, the variability between the bins seems very high - every single sgRNA spans effect sizes from +2 to -5 or -6 for MYC?
- Figure 1d caption: Change "ass" to "as"
- There is no mention of depositing the barcoded cloning vector. This should be made available on Addgene. Also, I could not find the publicly available GitHub code. Please include a URL.

Reviewer #2:

The authors of this study present a novel approach to trace single guide RNAs in pooled CRISPR screens by integrating random sequence labels (barcodes) into a guide RNA library. The authors claim that this strategy improves both the precision and accuracy of CRISPR screens, with low "costs" on the number of cells and sequencing reads - compared to screens using guide RNA libraries without random sequence labels. They test their approach in a pooled CRISPR screen in a colorectal cancer cell line and highlight the ability of this approach to score "internal replicates" (replicates of the same guide RNA from different cell lineages / populations) and to perform dropout analysis per lineage (per random sequence label of the same guide RNA). Both types of analyses outperform a total count analysis, which is used for most published pooled CRISPR screens using guide RNA libraries without random sequence labels.

Pooled CRISPR screens are a major tool for loss and gain of function screening in many labs, and approaches to further improve this method with regards to reproducibility, resolution, specificity, efficiency, and/or applicability will very likely have a huge impact on progressing science. The study presented by Schmierer and Botla et al. aims at improving the resolution of guide RNA reads by integrating an additional barcode (random sequence label) to each guide - in this regard the method is a technical advance of the current CRISPR screening technology. While barcoding approaches have been used previously for tracing cell lineages, this method has not been applied in the context of pooled CRISPR screening (double barcoding). It would provide a better resolution on the effects of single guides, specifically in the context of more focused CRISPR screens (e.g. functional subsets), where the resolution of single cells / cell populations is needed for a better understanding of the variability of the phenotype. However, it might not be scalable to a genome scale, or at least not be practicable, due to the increased number of cells and the sequencing depth needed. But with its increased resolution on sub-genome screens the method would be of broad interest for the CRISPR screening community.

Major points:

This manuscript is probably supposed to be a short application note with 2 Figures. I understand the rationale the authors use to present the application, but think - specifically considering the depth of data that was generated - that some of the panels could have a little more content. Figure 1d shows a nice example comparing internal replicates for MYCN vs. MYC across different guides. I think the "negative control guides" panel is redundant and in that sense and could be moved to the supplement (or could be used instead of the MYCN, to keep panels d) and e) consistent). Instead, one of the supplemental panels (from S1c) could be put into the main figure to provide an overview of the entire dataset, and to harmonize it with the LDA plot in Figure 1e. With regards to harmonization, it would be great to see MYC and "positive controls" (I assume ribosomal proteins - please state in Figure legend) labelled in both scatter plots, for the readers to visually compare their "location" in

the context of the full dataset.

When the authors compare IRA and LDA to TCA (Figure 2a), could they please state to which extend the results of IRA and LDA are similar (e.g. correlated, or similar rankings)? I understand that in different assays IRA and LDA could be very different, but they look very similar for the screen performed in this study (and maybe "redundant") - but I understand the authors want to make the point that both approaches outperform TCA.

In Figure 2c, why is this analysis not showing results for IRA? Could the authors also comment on the huge error bar for the 1/4 screen size (the error bar for LDA is as big as the bar itself). This comparison doesn't look significant (instead the authors use words like "massively" or "dramatically" increased), which would argue that it might be challenging to get statistically robust calls with small "screen sizes" / "sequencing depth" in genome-scale studies (see also my comment on the discussion below).

Minor points:

The discussion is sound, but the authors should also indicate the potential limitations of their approach. Specifically, the results in Figure 2c indicate that it might be challenging to get sufficient statistical power for genome-scale studies without sufficient sequencing depth / number of cells. The authors mention that they see the approach "instrumental in the interrogation of small genomic features" (lines 85-86), but they should also mention whether they see it applicable or not for genome-scale studies.

Further, the authors could put their study in the context of similar approaches and highlight advantages / disadvantages. E.g. there was a recent paper from Kalhor et al. ("Rapidly evolving homing CRISPR barcodes", Nature Methods, 2016), which presents a method that could also be used for lineage tracing and cellular barcoding - also its application for pooled CRISPR screening might not be as straightforward. Another approach to increase resolution on pooled CRISPR screens was published by Datlinger et al. ("Pooled CRISPR screening with single-cell transcriptome readout", Nature Methods, 2017), which could be briefly mentioned in the context of this manuscript.

The outline of the approach is very clear and the setup of the pooled screen is very robust with regards to sequencing depth, number of guides per gene, and choice of target genes (including negative and positive controls), to enable sufficient benchmarking. While most of the experimental methods are described well and detailed, I am missing some details about the library construction and computational methods.

Could the authors briefly describe how the random sequence labels were implemented in the cloning strategy - instead of just referring to the original paper that introduced unique molecular identifiers in a different context (Kivioja et al., 2012)? And could they also provide a histogram in the supplement showing the distribution of RSLs per guide RNA in the library?

A minor detail: The methods section also doesn't explicitly state how many replicates of the screen (in RKO cells) were performed - I assume 2?

Further, the analysis scripts mentioned in the methods should be made available online (e.g. as mentioned by the authors, via Github) before the publication of the manuscript, otherwise these sections don't have any meaning.

The computational analysis part needs some clarification: the description of the analysis (lines 164 to 218) should be put together in a way that it is clear to the reader what was done step-by-step. The different parts seem disconnected, which makes it also very difficult to understand what went into the data displays in Figures 1d-e and 2.

It is also not very clear from the methods how the guide RNAs were summarized per gene (e.g. in Figure 2a, each point represents a gene, that was targeted by ~10 guide RNAs with many RSLs - how was the effect of guide RNAs summarized on a gene level?)

Conclusion:

Despite my major points - that could all be addressed by working on the manuscript - I think this manuscript would be very valuable for the CRISPR screening community if published.

General response to reviewers and editorial comments

We are pleased to see that both reviewers consider our manuscript an important improvement to the current state of the art pooled CRISPR/Cas9 screening technology, and are grateful for their insightful comments and suggestions, which we address point by point in this rebuttal. We have made amendments to the manuscript to accommodate the suggestions (highlighted in red in the main text). We have also included several new Figure panels, and increased the number of Figures from two to three.

Response to Reviewer#1:

Schmierer et al. present a nice twist on CRISPR screens. Although previous work (e.g. Perturb-Seq papers from Regev and Weissman labs) already have incorporated UMIs in their pooled CRISPR screens, I think a paper emphasizing the advantages of UMIs with a careful comparison to non-UMI screens will be useful for the field.

We are pleased that the reviewer appreciates the importance of our UMI method. Perturb-Seq does use UMI-barcoding, however these UMIs are used as a proxy for the guide sequence, which would otherwise be difficult to obtain in single-cell RNASeq experiments. This is an entirely different application, in which UMIs serve a very different purpose. We use our RSLs for lineage tracing. We have now added a short discussion of how our approach relates to other techniques using UMIs and have cited the relevant papers (page 2, line 32-36).

1. Line 62 mentions binning of 64 replicates (barcodes per guide) but Fig. 1b indicates 4 bins under IRA.

The binning shown in old Figure 1b was just an example showing binning into 4 internal replicates, however any number of bins can be used. In the actual analysis, 64 internal replicates were used. To make this clearer, we have now indicated in the figure (now Figure 1c) that the 4 bins shown are just an example. We have also made this clearer in the figure legend (page 12, line 401).

2. TracrRNA in Figure 1a is incorrect. Please re-label as sgRNA.

Has been relabeled.

3. It would be great to get a better sense of the variability (I assume it is with respect to effect size) between the IRA bins. Could the authors show data on the variability of these replicates in a few different example genes? And summarize the variability across all genes in the library?

We have added a new Supplementary Figure (Fig. EV4) showing MYC and two additional example genes and the variability in effect size in different binning strategies (4, 16 and 64 internal replicates). The figure also shows a boxplot of the overall variability (MAD, median absolute deviation) across all guides in the library.

4. For Figure 1d, it is unclear what exactly is being plotted. Does each curved line contain 64 replicate bins? If so, the dots must be plotted with alpha shading. Otherwise, it is difficult to see where the bins concentrate. Also, based on these plots, the variability between the bins seems very high - every single sgRNA spans effect sizes from +2 to -5 or -6 for MYC?

Old Figure 1d (now Figure 2a) shows the variability of the IRA bins for two example genes and negative control guides. Each curved line contains 64 bins, and we now have modified the Figure such that the dots are now plotted with alpha-shading as the reviewer suggests.

The variability between bins is indeed relatively high and is likely due to several factors:

- a) **Variation in Cas9 and guide expression.** Cells carrying the same guide but being derived from distinct cell lineages will vary in the expression levels of both guide and Cas9, e.g. depending on where in the genome the transgenes integrated. sgRNA and Cas9 stoichiometry can heavily influence the kinetics of cutting (see for instance Wright AV et al., PNAS 2015)
- b) **Distinct repair outcomes.** In frame repair can leave the gene product functional or partly functional.
- c) **Variation in cell growth characteristics.** Due to its exponential nature, cell proliferation is the main determinant of enrichment/relative depletion in these types of screens, and cell-to-cell variability in cell cycle length (depending for instance on the microenvironment in the plate) will introduce substantial variability.

The heterogeneity observed here occurs whether or not RSLs are present, however remains undetectable in the traditional approach. It is important to note that the high variability at the guide level does not preclude calling of a large number of significant hits at the gene level. We have now included a brief discussion of these issues in the main text (page 3, line 67-72).

5. Figure 1d caption: Change "ass" to "as"

We apologize for the typo, this has been changed.

6. There is no mention of depositing the barcoded cloning vector. This should be made available on Addgene. Also, I could not find the publicly available GitHub code. Please include a URL.

The barcodes come from an oligo that is cloned together with the guide sequences, thus there is no barcoded vector. We have now explained the cloning strategy in much greater detail in the Method section (page 4, line 113 and on). We have also included a new panel in Figure 1a to make clear how the RSL-library was assembled. We will of course make the parental vector available, so the method can be easily reproduced. All the scripts used along with a document for their usage will be uploaded to GitHub and will be made publicly available once the manuscript is published. The link is now given in the new section "data and software availability".

Reviewer#2:

The authors of this study present a novel approach to trace single guide RNAs in pooled CRISPR screens by integrating random sequence labels (barcodes) into a guide RNA library. The authors claim that this strategy improves both the precision and accuracy of CRISPR screens, with low "costs" on the number of cells and sequencing reads - compared to screens using guide RNA libraries without random sequence labels. They test their approach in a pooled CRISPR screen in a colorectal cancer cell line and highlight the ability of this approach to score "internal replicates" (replicates of the same guide RNA from different cell lineages / populations) and to perform dropout analysis per lineage (per random sequence label of the same guide RNA). Both types of analyses outperform a total count analysis, which is used for most published pooled CRISPR screens using guide RNA libraries without random sequence labels.

Pooled CRISPR screens are a major tool for loss and gain of function screening in many labs, and approaches to further improve this method with regards to reproducibility, resolution, specificity, efficiency, and/or applicability will very likely have a huge impact on progressing science. The study presented by Schmierer and Botla et al. aims at improving the resolution of guide RNA reads by integrating an additional barcode (random sequence label) to each guide - in this regard the method is a technical advance of the current CRISPR screening technology. While barcoding approaches have been used previously for tracing cell lineages, this method has not been applied in the context of pooled CRISPR screening (double barcoding). It would provide a better resolution on the effects of single guides, specifically in the context of more focused CRISPR screens (e.g. functional subsets), where the resolution of single cells / cell populations is needed for a better understanding of the variability of the phenotype.

However, it might not be scalable to a genome scale, or at least not be practicable, due to the increased number of cells and the sequencing depth needed. But with its increased resolution on sub-genome screens the method would be of broad interest for the CRISPR screening community.

We are pleased that the reviewer thinks that our improvement to CRISPR/Cas9 screening is likely to have a big impact on progressing science. The Reviewer seems to be skeptical whether the method is applicable in genome-wide screens, however, as shown in Figure 3c, our method allows to determine statistically significant hit genes at a much lower screen size (cells per guide) and sequencing depth. This is perhaps counterintuitive, but because any screen needs to use relatively large number of cells per guide to achieve statistical power, tagging each individual lineage incurs no cost but increases the amount of information that is obtained from the same number of cells, consequently improving both precision and accuracy of the screen at any given screen size. In other words, whereas traditional screen obtains only the sum of reads derived from all cells containing a particular guide, the UMI design obtains the read counts for each individual lineage. The resulting distribution can then be analyzed to improve the statistical power. We have added a new Enhanced View Figure, Fig. EV1 to make this clear. Thus, genome-scale

screens will benefit from RSLs just as smaller screens, and we are currently in the process of testing this thoroughly. We have also explicitly stated this in the summary (page 4, line 109)

Major points:

This manuscript is probably supposed to be a short application note with 2 Figures. I understand the rationale the authors use to present the application, but think - specifically considering the depth of data that was generated - that some of the panels could have a little more content.

We have expanded and rearranged the Figures (see also response to point 1 below). An additional Figure panel has been moved from the supplement into the main figure (now Figure 1b), and the number of Figures has been increased from two to three.

1. Figure 1d shows a nice example comparing internal replicates for MYCN vs. MYC across different guides. I think the "negative control guides" panel is redundant and in that sense and could be moved to the supplement (or could be used instead of the MYCN, to keep panels d) and e) consistent). Instead, one of the supplemental panels (from S1c) could be put into the main figure to provide an overview of the entire dataset, and to harmonize it with the LDA plot in Figure 1e. With regards to harmonization, it would be great to see MYC and "positive controls" (I assume ribosomal proteins - please state in Figure legend) labelled in both scatter plots, for the readers to visually compare their "location" in the context of the full dataset.

We have considered the reviewer's suggestion regarding old Figure 1d (now Figure 2a), however since Reviewer 1 wanted to see more examples rather than less, we have decided to leave the figure as it is. We have however included an additional panel (now 2b) showing the replicate correlation of IRA/SSMD scores, to allow direct comparison between IRA/SSMD and LDA. We have also harmonized the Figures as suggested, and highlighted MYC as well as the positive controls (which are now clearly labelled as ribosomal proteins).

2. When the authors compare IRA and LDA to TCA (Figure 2a), could they please state to which extent the results of IRA and LDA are similar (e.g. correlated, or similar rankings)? I understand that in different assays IRA and LDA could be very different, but they look very similar for the screen performed in this study (and maybe "redundant") - but I understand the authors want to make the point that both approaches outperform TCA.

Indeed both methods outperform TCA, and perform in a very similar way. To show this, we include a Figure for the Reviewer's inspection (Figure for Reviewer 2). The figure shows the correlation between the gene ranks obtained by the LDA and IRA/SSMD methods. As also

seen from what is now Fig 3a, the reviewer's expectation of a strong correlation between the gene ranks obtained by both methods is correct. This correlation is robust to the screen size, at least in the range tested here. We do not think that the two methodologies are redundant, but that either one or the other might be superior depending on the specific parameters of a screen (sequencing depth, number of cells per guide, number of RSLs per guide, etc). This is now explained on page 4, lines 90-93.

In Figure 2c, why is this analysis not showing results for IRA? Could the authors also comment on the huge error bar for the 1/4 screen size (the error bar for LDA is as big as the bar itself). This comparison doesn't look significant (instead the authors use words like "massively" or "dramatically" increased), which would argue that it might be challenging to get statistically robust calls with small "screen sizes" / "sequencing depth" in genome-scale studies (see also my comment on the discussion below).

We are grateful to the reviewer for pointing out this inconsistency. We have now included also IRA/SSMD in the figure. We also realized that we had analyzed the data for the LDA with an outdated version of the MAGeCK software package, (Version 0.5.3). Briefly, we drew up a ranked guide list (guides that lost most RSLs during the screen with the lowest rank) and used the RRA algorithm as implemented in the MAGeCK software package to call significantly depleted genes from the ranked guide-list. Re-analysis with MAGeCK version 0.5.6, which had several bugs fixed, yielded a much higher number of genes depleted at 1% FDR, and consequently much more consistent results. To corroborate this improvement, we also analyzed the data with a different statistical tool which gave very similar results. As a consequence, the large error bar for LDA is now much smaller. We are in contact with the originators of the MAGeCK software package regarding this issue. The differences in output of different versions of sophisticated software tools also highlights the benefit of the UMI approach, as it allows analysis of results using multiple types of simple and standard statistical tools, which, unlike most software, can be proven to give correct results.

Minor points:

3. The discussion is sound, but the authors should also indicate the potential limitations of their approach. Specifically, the results in Figure 2c indicate that it might be challenging to get sufficient statistical power for genome-scale studies without sufficient sequencing depth / number of cells. The authors mention that they see the approach "instrumental in the interrogation of small genomic features" (lines 85-86), but they should also mention whether they see it applicable or not for genome-scale studies.

Figure 3c (previously Fig 2c) makes now very clear that the presence of RSLs allows to downsize the screen, and still obtain a much larger number of statistically significant hits than with the conventional method. Inclusion of RSLs can thus push

the lower limit of cells per guide and number of reads required. We have no doubt that also genome-wide screens will benefit considerably from the inclusion of RSLs, with only marginally higher cost. We have included Fig EV1, which shows in cartoon form that RSLs do not require larger cell numbers, but give more information from an identical experimental setup.

4. Further, the authors could put their study in the context of similar approaches and highlight advantages / disadvantages. E.g. there was a recent paper from Kalhor et al. ("Rapidly evolving homing CRISPR barcodes", Nature Methods, 2016), which presents a method that could also be used for lineage tracing and cellular barcoding - also its application for pooled CRISPR screening might not be as straightforward. Another approach to increase resolution on pooled CRISPR screens was published by Datlinger et al. ("Pooled CRISPR screening with single-cell transcriptome readout", Nature Methods, 2017), which could be briefly mentioned in the context of this manuscript.

We have now added a short discussion of how our approach relates to similar techniques and have cited the relevant papers (page 2, line 32-36). However, we believe that the homing barcode approach in its current form would not work well in dropout screens, as cutting DNA slows down cell division, increasing the variability of the assay.

5. The outline of the approach is very clear and the setup of the pooled screen is very robust with regards to sequencing depth, number of guides per gene, and choice of target genes (including negative and positive controls), to enable sufficient benchmarking. While most of the experimental methods are described well and detailed, I am missing some details about the library construction and computational methods. Could the authors briefly describe how the random sequence labels were implemented in the cloning strategy - instead of just referring to the original paper that introduced unique molecular identifiers in a different context (Kivioja et al., 2012)?

We have now explained how the RSLs are cloned together with the guide (Page 4 and on, section "oligo synthesis and library cloning").

6. And could they also provide a histogram in the supplement showing the distribution of RSLs per guide RNA in the library?

The distribution of RSLs per guide in the library can be seen from the boxplot in Supplementary Figure EV2.

7. The methods section also doesn't explicitly state how many replicates of the screen (in RKO cells) were performed - I assume 2?

Yes, the RSL library was screened in the RKO cells in two replicates. This is now mentioned (page 6, line 168).

8. Further, the analysis scripts mentioned in the methods should be made available online (e.g. as mentioned by the authors, via Github) before the publication of the manuscript, otherwise these sections don't have any meaning.

All the scripts used along with a document for their usage will be uploaded to GitHub and will be made publicly available once the manuscript is published. The link is now given in the new section “data and software availability”.

The computational analysis part needs some clarification: the description of the analysis (lines 164 to 218) should be put together in a way that it is clear to the reader what was done step-by-step. The different parts seem disconnected, which makes it also very difficult to understand what went into the data displays in Figures 1d-e and 2. It is also not very clear from the methods how the guide RNAs were summarized per gene (e.g. in Figure 2a, each point represents a gene, that was targeted by ~10 guide RNAs with many RSLs - how was the effect of guide RNAs summarized on a gene level?)

We have re-structured the data analysis section of the online methods (page 7, line 209 and on), to clarify this. We have also amended figure legends to include important details (page 11, lines 422-426 and line 429).

Thank you again for sending us your revised manuscript. We are now satisfied with the modifications made and I am pleased to inform you that your paper has been accepted for publication.

Corresponding Author Name: Prof. Jussi Taipale

Manuscript Number: MSB-17-7834